# RDC complex executes a dynamic piRNA program during *Drosophila* spermatogenesis to safeguard male fertility

**Peiwei Chen**⊙*, **Yicheng Luo**⊙, **Alexei A. Aravin**⊙*

California Institute of Technology, Division of Biology and Biological Engineering, Pasadena, California, United States of America

* peiweitc@gmail.com (PC); aaa@caltech.edu (AAA)

**Data Availability Statement:** All data are available within the manuscript except for the sequencing data, which is available at NCBI SRA (accession number: PRJNA646006).

## Abstract

piRNAs are small non-coding RNAs that guide the silencing of transposons and other targets in animal gonads. In *Drosophila* female germline, many piRNA source loci dubbed "piRNA clusters" lack hallmarks of active genes and exploit an alternative path for transcription, which relies on the Rhino-Deadlock-Cutoff (RDC) complex. RDC was thought to be absent in testis, so it remains to date unknown how piRNA cluster transcription is regulated in the male germline. We found that components of RDC complex are expressed in male germ cells during early spermatogenesis, from germline stem cells (GSCs) to early spermatocytes. RDC is essential for expression of dual-strand piRNA clusters and transposon silencing in testis; however, it is dispensable for expression of Y-linked *Suppressor of Stellate* piRNAs and therefore *Stellate* silencing. Despite intact *Stellate* repression, males lacking RDC exhibited compromised fertility accompanied by germline DNA damage and GSC loss. Thus, piRNA-guided repression is essential for normal spermatogenesis beyond *Stellate* silencing. While RDC associates with multiple piRNA clusters in GSCs and early spermatogonia, its localization changes in later stages as RDC concentrates on a single X-linked locus, *AT-chX*. Dynamic RDC localization is paralleled by changes in piRNA cluster expression, indicating that RDC executes a fluid piRNA program during different stages of spermatogenesis. These results disprove the common belief that RDC is dispensable for piRNA biogenesis in testis and uncover the unexpected, sexually dimorphic and dynamic behavior of a core piRNA pathway machinery.

## Author summary

Large fractions of eukaryotic genomes are occupied by mobile genetic elements called transposons. Active transposons can move in the genome causing DNA damage and mutations, while inactive copies can contribute to chromosome organization and regulation of gene expression. Host cells employ several mechanisms to discriminate transposons from other genes and repress transposon activities. In germ cells, a conserved class of short RNAs called Piwi-interacting (pi)RNAs recognize target RNAs in both the nucleus and cytoplasm and then guide transposon repression by preventing their transcription

**Funding:** This work was supported by the Howard Hughes Medical Institute Faculty Scholar Award (https://www.hhmi.org) and the National Institutes of Health R01 GM097363 (https://www.nih.gov) to AAA. The funders had no role in study design, data collection and analysis, decision to publish, or preparation of the manuscript.

**Competing interests:** The authors have declared that no competing interests exist.

and destroying their RNAs. piRNAs are encoded in extended genomic regions dubbed piRNA clusters. Previously, composition and regulation of piRNA clusters were studied in the female germline of fruit flies, where a nuclear protein complex, the RDC complex, was shown to promote non-canonical transcription of these regions. However, RDC was believed to be dispensable in males. Here, we showed that RDC is essential for transposon repression in males, and males lacking RDC exhibit compromised fertility and loss of germ cells. We found that RDC binds multiple piRNA clusters in early germ cells but concentrates on a single locus at later stages. Our results indicate dynamic regulation of loci that produce piRNAs and, therefore, piRNA targets throughout spermatogenesis.

## Introduction

Transposable elements (TEs) are selfish genetic elements that have the ability to propagate in the genome. When unchecked, transposition of TEs can cause overwhelming DNA damage and, eventually, genome instability. This poses a particular threat to germ cells, and TE de-repression often leads to reproductive defects like sterility. To cope with this, a small RNA-mediated genome defense mechanism involving PIWI proteins and PIWI-interacting RNAs (piRNAs) is employed in animal gonads to silence TEs [1].

In ovaries of *Drosophila melanogaster*, most piRNAs are made from so-called dual-strand piRNA clusters, where both genomic strands are transcribed to give rise to piRNA precursors. Transcription of dual-strand piRNA clusters is unusual in a number of ways. First, there is no clear promoter signature for transcription initiation [2]. Second, splicing, termination and poly-adenylation of nascent transcripts are all suppressed [2–4]. Third, transcription occurs at the presence of H3K9me3 [5], a histone modification generally seen as a repressive mark for gene expression. In fact, canonical transcription must be repressed by factors like Maelstrom to allow proper piRNA production from dual-strand piRNA clusters [6]. Prior work has shown that such non-canonical transcription and co-transcriptional processing of piRNA precursors depend on the RDC complex, composed of Rhino (Rhi), Deadlock (Del) and Cutoff (Cuff) proteins, that bind dual-strand clusters [2,4]. Rhi belongs to the HP1 family and binds H3K9me3 through its chromo-domain, anchoring the RDC complex onto dual-strand clusters [2,4,7]. Cuff is a homolog of the conserved cap-binding protein Rai1 that was reported to suppress both splicing [4] and transcriptional termination [3], in order to facilitate the production of long, unspliced piRNA precursors. Del, on the other hand, recruits a paralog of transcription initiation factor TFIIA-L, Moonshiner (Moon), to initiate transcription in hostile heterochromatin environment [8]. Together, the RDC complex conveys transcriptional competence to dual-strand piRNA clusters, and the majority of piRNA production collapses when one of its components is missing.

While piRNA pathway is known to be active in both male and female germline of *Drosophila*, expression and functions of Rhi, Del and Cuff were studied exclusively during oogenesis. Mutations of *rhi*, *del* and *cuff* were shown to cause female sterility, however, mutant males remained fertile [9,10]. In addition, *rhi*, *del* and *cuff* are predominantly expressed in ovaries, with low or no expression in testes and somatic tissues [11–13]. Similarly, Moon was also believed to be ovary-specific, given that a high expression level could only be found in ovaries [8]. Finally, the silencing of *Stellate* by abundant *Suppressor of Stellate* piRNAs was shown to be unperturbed in testes of *rhi* and *cuff* mutants, suggesting that the RDC complex is dispensable for piRNA biogenesis in males [14,15]. Collectively, these findings led to the notion that

RDC complex is dispensable for piRNA pathway in the male germline, raising the question of how piRNA cluster expression is controlled during spermatogenesis.

Here, we describe a developmentally regulated assembly of RDC complex in testes. We found that low expression of RDC complex components can be attributed to the fact that only a small subset of cells at early stages of spermatogenesis express *rhi*, *del* and *cuff*. Loss of RDC complex in testes results in a collapse of piRNA production, TE de-repression, and, ultimately, compromised male fertility, supporting an indispensable role of RDC complex in spermatogenesis. Even though RDC complex is assembled and functional in both sexes, we found differential genome occupancies of RDC complex between two sexes, correlating with sexually dimorphic usage of genome-wide piRNA source loci. Finally, RDC complex appears to exhibit dynamic binding on different piRNA clusters during spermatogenesis, allowing different piRNAs source loci to be used at different stages of early sperm development.

## Results

### Components of the RDC complex are required for normal male fertility

Previous studies showed that, while *rhi* is required for female fertility, it is dispensable for male fertility [13,14]. In agreement with this, we found that *rhi* mutant males indeed produce progeny when crossed with wildtype females. However, careful examination of the male fertility by sperm exhaustion test [16] revealed significantly compromised fertility in *rhi* mutant males. Even though most *rhi* mutant males were initially fertile, the percentage of fertile males dropped as they aged and stayed low from day 3 in comparison to heterozygous sibling controls (Fig 1A). To probe male fertility more quantitatively, we repeated the test and counted numbers of progeny for each male every day. We found that even young, 1-day-old *rhi* mutant males, which were fertile, produced fewer progeny than heterozygous sibling controls (Fig 1A). Also, *rhi* mutant males produced nearly no progeny after two-day sperm exhaustion, while heterozygous sibling controls continued to produce ~100 progeny on average throughout the sperm exhaustion process. To extend this observation to other components of the RDC complex, we repeated the sperm exhaustion tests for *del* and *cuff* mutant males. Both *del* and *cuff* mutants displayed a reduction in male fertility compared to their respective heterozygous sibling controls (n = 5, $P \leq 0.001$, S1 Fig) These results demonstrate that male fertility is substantially compromised at the absence of *rhi*, *del* or *cuff*, suggesting an indispensable role of all three RDC complex components in maintaining normal male fertility.

### RDC complex is assembled in nuclei of germ cells from GSCs to early spermatocytes

The dependency of normal male fertility on *rhi*, *del* and *cuff* prompted us to re-examine whether RDC complex is assembled in testis. modENCODE data and previous work showed that tissue-wide mRNA levels of *rhi*, *del* and *cuff* are high in ovaries but low in testes and the soma [11–13], which led to the notion that RDC might be ovary-specific [12,13,17]. To examine expression of Rhi, Del and Cuff in testis, we took an imaging-based approach that provides single-cell resolution and preserves spatial information. We examined expression of individual components of the RDC complex using GFP-tagged Rhi, Del and Cuff transgenes that are expressed under the control of their native regulatory regions [2]. Importantly, GFP-tagged RDC components are functional, as their expression fully rescued the sterility of the respective female mutant (S2 Fig). All three proteins are expressed at the apical tip of testis that contains germ cells at early steps of spermatogenesis (Fig 1B and 1C), indicating that all three components of RDC complex are expressed in testis, though only in a small subset of the cells.

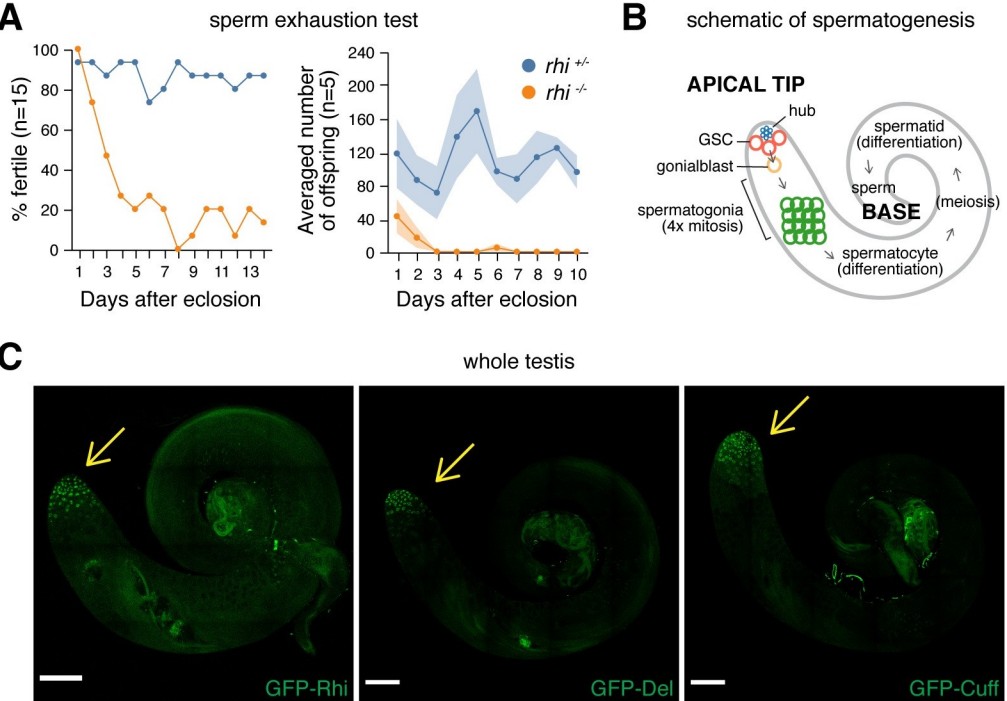

**Fig 1. Rhi is required for normal male fertility, and components of RDC complex are expressed at the apical tip of testis.** (A) Compromised fertility *of rhi* mutant males. Sperm exhaustion test of *rhi^2/KG* mutant (orange) and heterozygous sibling control (blue) males. Left: percentages of fertile males 1–14 days post-eclosion (n = 15). Right: averaged numbers of offspring per male 1–10 days after eclosion (n = 5). Shaded areas display standard error. Two charts report results from two independent sperm exhaustion tests. (B) A schematic of spermatogenesis, showing major developmental stages of male germline as well as the somatic hub that serves as germline stem cell (GSC) niche. (C) Expression of GFP-tagged Rhi (left), Del (middle) and Cuff (right) transgenes driven by their respective regulatory regions. Expression of all three proteins can only be seen at the apical tip of testis (pointed to by the yellow arrow). Scale bar: 100µm.

Rhi, Del and Cuff form foci in nuclei (Fig 2A). To test whether all three proteins co-localize in nuclear foci, we tagged Rhi with a different fluorophore and expressed it using the previously described regulatory region of *rhi* [4]. After verifying that this transgene rescues female sterility of the *rhi* mutation (S2 Fig), we analyzed its localization with Del and Cuff. Indeed, Rhi co-localizes with both Del and Cuff (Fig 2A), consistent with the formation of RDC complex. Next, we examined the inter-dependence of Rhi, Del and Cuff localization (Fig 2B). In either *del* or *cuff* mutants, Rhi becomes dispersed and no longer forms puncta in nuclei. Similarly, Del also disperses at the absence of Rhi or Cuff. Expression of Cuff is strongly decreased in both *rhi* and *del* mutants, indicating its destabilization. Therefore, Rhi, Del and Cuff co-localize in distinct nuclear foci that depend on the simultaneous presence of all three proteins.

We further characterized the expression of RDC complex in testis. Rhi, Del and Cuff are expressed in nuclei of germline stem cells (GSCs) that are directly adjacent to somatic hub cells labeled by Fas3, but not in hub cells (Fig 2C). Rhi expression continues beyond spermatogonia marked by Bam, until early spermatocytes that express Sa (Spermatocyte arrest) (Figs 2D and S3). In *bam* mutant testes, where spermatogenesis is arrested at the spermatogonia-to-spermatocyte transition stage, we observed an expansion of spermatogonia and expression of Rhi throughout entire testes (Fig 2E). In addition to germ cells and hub cells, the apical tip of testes contains somatic gonadal cells (cyst stem cells and early cyst cells) that can be distinguished from germ cells by Tj expression. Rhi is not expressed in somatic cells that express Tj

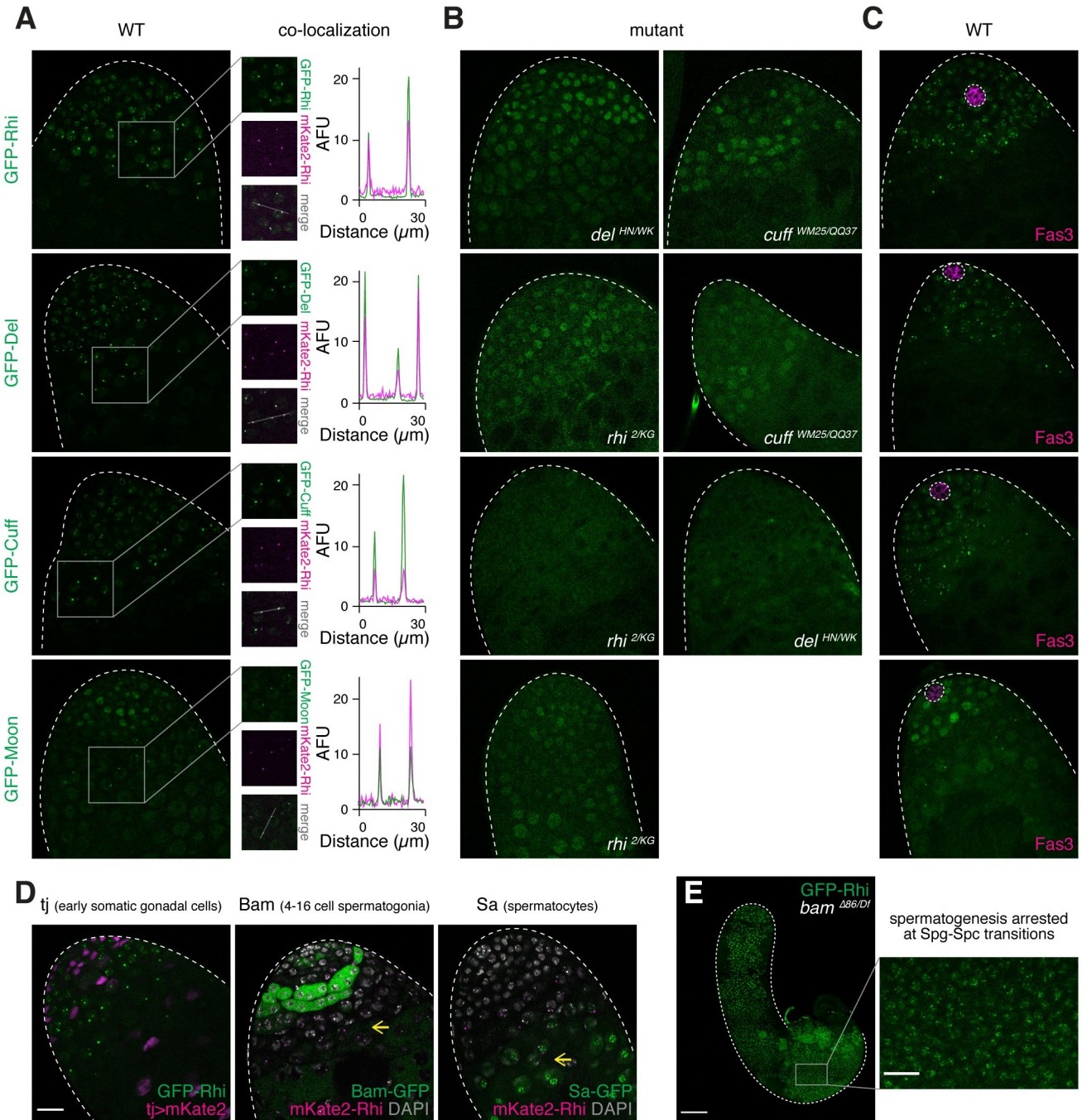

**Fig 2. RDC complex is assembled in early male germ cells.** (A) Rhi, Del, Cuff and Moon co-localize in nuclear foci. Confocal images showing apical tips of testes expressing GFP-tagged Rhi, Del, Cuff and Moon (top to bottom) transgenes driven by their native regulatory regions. Co-localization with mKate2-tagged Rhi in nuclear foci are shown on the right. Signal intensities along the marked line are plotted for each of four co-localization analysis. AFU, arbitrary fluorescence units. (B) Inter-dependence of Rhi, Del and Cuff localization in nuclear foci, as well as the dependence of Moon localization on Rhi. Confocal images showing apical tips of testes expressing GFP-Rhi in $del^{HN/WK}$ and $cuff^{WM25/QQ37}$, GFP-Del in $rhi^{2/KG}$ and $cuff^{WM25/QQ37}$, GFP-Cuff in $rhi^{2/KG}$ and $del^{HN/WK}$, and GFP-Moon in $rhi^{2/KG}$ mutant backgrounds. Nuclear foci of each protein dispersed or disappeared in respective mutants. (C) Rhi, Del, Cuff and Moon are expressed in GSCs. Immuno-fluorescence of testes expressing GFP-tagged Rhi, Del, Cuff and Moon, stained for somatic hub marker Fas3. Note that GSCs directly adjacent to Fas3-positive hub express all four proteins. (D) Rhi is not expressed in somatic gonadal cells, and its germline expression continues beyond spermatogonia till early spermatocytes. Confocal images showing a fluorescently tagged Rhi transgene with somatic gonadal cells marked by $tj$-$Gal4$>$UASp$-$mKate2$ (left), 4–16 cell spermatogonia marked by Bam-GFP (middle) and spermatocytes marked by Sa-GFP (right). Expression of Rhi in early spermatocytes is pointed to by yellow arrows. (E) Rhi expression upon arrest of spermatogenesis in $bam$ mutants. Confocal image showing $bam^{Δ86/Df}$ mutant testis expressing GFP-Rhi, where spermatogenesis is arrested at the spermatogonia-to-spermatocyte transition stage. Note

that spermatogonia are expanded and virtually all germ cells express Rhi. An enlarged view of the basal part of mutant testis is shown at the bottom, which is on the same scale as (D). All images share scale bars with (D), except for (E). Scale bars: 20μm (D) and 100μm (E).

(Figs 2D and S3), confirming its restriction to the germline. Taken together, we conclude that RDC complex is assembled in male germ cells during spermatogenesis, from GSCs to spermatogonia and early spermatocytes.

In ovaries, RDC complex is known to promote piRNA cluster transcription by two mechanisms: 1) suppression of premature transcriptional termination, a function mediated by Cuff [3], and 2) licensing of non-canonical transcriptional initiation, a function that requires the recruitment of a basal transcriptional factor TFIIA-L paralog, Moonshiner (Moon) [8]. Expression of Moon was reported to be specific to female germline, raising the question of whether RDC complex can fulfill its function in the male germline, if its functional partner is missing. However, we observed Moon expression in testis using a GFP-tagged Moon transgene (expressed under its native regulatory region) that is able to rescue the female sterility caused by the *moon* mutation [8] (Fig 2A). Importantly, Moon is expressed at similar stages as components of RDC complex, from GSCs to spermatogonia and early spermatocytes at the apical tip of testis (Fig 2A and 2C). Furthermore, Moon co-localizes with Rhi in nuclear foci from late spermatogonia to early spermatocyte, and its focal localization is abolished in *rhi* mutants (Fig 2A and 2B). In GSCs, Moon localization is more diffused than components of the RDC complex (Fig 2A and 2C). However, Moon expression is perturbed in *rhi* mutants, even in GSCs (Fig 2B), suggesting that Moon depends on Rhi for protein stability and proper localization in the nucleus throughout its expression window. On the contrary, Rhi localization appears normal in two different *moon* mutants, $moon^{\Delta 1}$ and $moon^{\Delta 28}$, indicating that Moon acts genetically downstream of RDC complex (S4 Fig). These observations suggest that RDC complex can recruit Moon to license transcription initiation in the male germline.

## Loss of RDC complex causes DNA damage and germ cell death in testis

To identify cellular mechanisms underlying fertility decline in males lacking RDC complex, we examined the morphology of germ cells marked by Vasa-GFP in *rhi* mutants. Normally, Vasa-positive germ cells are tightly packed at the apical tip of testis, as we observed in testes of heterozygous control males (n = 162; Fig 3A). However, half of *rhi* mutant testes (54.9%, n = 134/244) had visibly fewer germ cells with prominent gaps in-between, indicative of an elevation of germ cell death (Fig 3A). Furthermore, another quarter of *rhi* mutant testes (25.4%, n = 62/244) completely lost early germ cells altogether, and only 19.7% (n = 48/244) showed wildtype-like germline morphology (Fig 3A and 3B). We concluded that loss of Rhi leads to a reduction in the germ cell count in testis.

Next, we examined impacts of Rhi loss on the resident germline stem cell (GSC) population. We quantified the number of GSCs per testis by counting the number of Vasa-positive germ cells directly adjacent in 3D to the somatic GSC niche labeled by Fas3. We found a reduction of GSCs in testes of *rhi* mutants compared with heterozygous controls in two age groups (1–4 and 9–12 days old) (P<0.0001, Mann–Whitney–Wilcoxon test, Fig 3C). About a quarter of *rhi* mutant testes did not have any GSC at all. Accordingly, we observed an expansion of Fas3-positive hub at a similar rate (24.4%, n = 59/244), which usually occurs at the absence of GSCs and is never seen in control testes (Fig 3A). Hence, GSC population sizes shrink drastically in testes lacking Rhi. Similar to *rhi* mutants, testes of 10-day-old *del* and *cuff* mutants are often completely depleted of early germ cells including GSCs (Fig 3D), indicating that loss of any component of the RDC complex leads to a collapse of spermatogenesis. Notably, aged virgin mutant males that lost all early germ cells, nevertheless, harbor mature sperm in their seminal

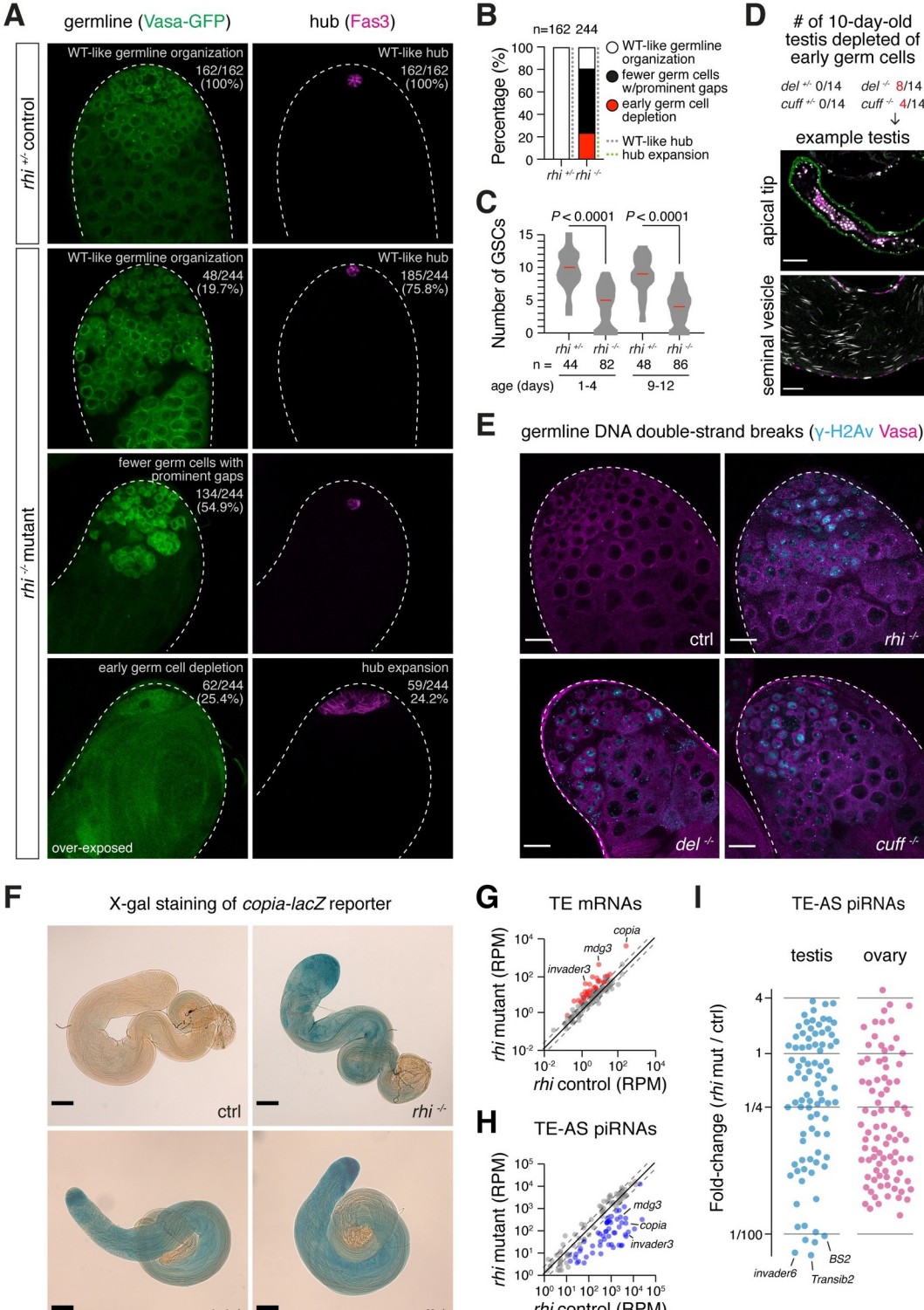

**Fig 3. Loss of Rhi causes germ cell death, DNA damage and TE de-repression.** (A) Loss of germ cells in testes of *rhi* mutants. Left: expression of germ cell marker, Vasa-GFP driven by *vasa* promoter, in testes of *rhi²/KG* mutants and heterozygous control. Right: somatic hub cells that form a niche for GSCs are marked with Fas3. Classification of germ cell phenotype (defined by number and organization of germ cells that express Vasa) and hub size phenotype (defined by cells that express Fas3) are labeled at the top right with corresponding statistics. All images on the same scale as (D). (B) Quantification

of germ cell survival and hub size in *rhi* heterozygous control (left) and mutant (right) testes shown in (A). n, number of testes examined. (C) Loss of GSCs in testes of *rhi* mutants. Violin plot showing GSC numbers in *rhi*$^{2/KG}$ mutant and age-matched heterozygous sibling control testes. Median of GSC number is marked red. GSC number is acquired by counting the total number of Vasa-positive cells directly adjacent to Fas3-positive hub in 3D. $P<0.0001$ based on Mann–Whitney–Wilcoxon test. n, number of testes counted for each genotype and age group. (D) Testes of *del* and *cuff* mutant males showed frequent loss of early germline but produce mature sperm indicating no block of spermatogenesis. Number of 10-day-old virgin males depleted of early germline is listed for each genotype at the top. The apical tip of testis and the seminal vesicle (SV) of such testes are shown below as an example, stained for Vasa, Fas3 and DAPI. Scale bar: 40μm and 20μm for testis and SV, respectively. (E) Accumulation of DNA double-strand breaks (DSBs) in male germline of *rhi*$^{2/KG}$, *del*$^{HN/WK}$ and *cuff*$^{WM25/QQ37}$ mutants. Immuno-fluorescence of heterozygous control and mutant testes, stained for γ-H2Av, a marker for DNA DSBs, and Vasa, a germline marker. Scale bars: 20μm. (F) De-repression of *copia* reporter in testes of *rhi* mutants. Brightfield images showing heterozygous control and *rhi*$^{2/KG}$, *del*$^{HN/WK}$ and *cuff*$^{WM25/QQ37}$ mutant testes expressing *copia-lacZ*, after X-gal staining. *copia* LTR containing its promoter is fused upstream to *lacZ* gene. Note that part of *copia* LTR is transcribed as well. Scale bar: 100μm. (G) De-repression of TEs in testes of *rhi* mutants measured by polyA+ RNA-seq. Scatter plot showing expression of TE mRNAs in *rhi*$^{2/KG}$ mutant versus heterozygous control testes. TEs that show ≥2-fold de-repression (FDR < 0.05) and ≥1 RPM averaged levels are marked red. The mean of two biological replicates is shown. (H) Loss of TE-targeting piRNAs in testes of *rhi* mutants. Scatter plot showing expression of TE-antisense piRNAs in *rhi*$^{2/KG}$ mutant versus heterozygous control testes. TE-antisense piRNAs that show ≥2-fold reduction (FDR < 0.05) and ≥10 RPM averaged levels are marked blue. Shown are averages of two biological replicates. (I) Loss of TE-targeting piRNAs in *rhi* males and females. Scatter plot showing fold-change of TE-antisense piRNAs upon loss of *rhi* in testis (left, blue) and ovary (right, pink). Note that piRNAs targeting several TE families demonstrate over 100-fold reduction in *rhi* testis, a magnitude not observed in ovary. Averages of two biological replicates are shown.

vesicles (Fig 3D), suggesting that disruption of the RDC complex does not block spermatogenesis at a specific stage in young males, but early germ cells are depleted when they age. Staining testes for the phosphorylated H2A variant (γ-H2Av), a marker for DNA double-strand breaks (DSBs), revealed massive accumulation of unrepaired DNA DSBs in early germ cells of *rhi*, *del* and *cuff* mutant testes (Fig 3E). Unrepaired DNA DSBs likely cause germ cell death in mutants of RDC complex components. Overall, our results suggest that the loss of early germ cells, including GSCs, accompanied by widespread unrepaired DNA DSBs is responsible for the compromised fertility of mutant males lacking an intact RDC complex.

## RDC complex is required for TE silencing in testis

Widespread DNA DSBs can result from TE transposition. To quantify TE expression, we sequenced polyadenylated (polyA+) RNAs from *rhi* mutant and control testes. PolyA+ RNA-seq revealed 32 TE families showing significant up-regulation in testes of *rhi* mutants (>2-fold increase, FDR < 0.05; Fig 3G). Among them, the most de-repressed ones include *mdg3* (36-fold), *invader3* (18-fold) and *copia* (13-fold). To verify TE de-repression, we employed a *copia-lacZ* reporter, where the long terminal repeat (LTR) of *copia* containing *copia* promoter is fused upstream to the *lacZ* gene and its expression can be directly examined by X-gal staining [18]. *copia* is known to be active in the male germline [19] and has the highest expression level among all TEs in testes [20]. Whereas no detectable X-gal staining was seen in control testes, robust staining was observed in *rhi* mutants (Fig 3F), confirming strong de-repression of *copia* at the absence of *rhi*. Similarly, the *copia* reporter was de-repressed in *del* and *cuff* mutant testes (Fig 3F). These results show that TEs are de-repressed in testes lacking a functional RDC complex.

In ovaries, RDC complex is required for piRNA production from dual-strand clusters to ensure efficient TE silencing [2–4,8]. To examine piRNA biogenesis, we sequenced and analyzed small RNAs in *rhi* mutant and control testes. We found a loss of antisense piRNAs targeting many TE families in *rhi* mutant testes (>2-fold reduction, FDR < 0.05; Fig 3H), suggesting an overall defect in piRNA production. In fact, there is a moderate correlation between the fold-derepression of TEs and the fold-reduction of TE-targeting piRNAs in testis upon mutating *rhi* (log-transformed values: Spearman's $\rho = 0.41$, $P = 7.3 \times 10^{-5}$, Pearson's $\rho =$

0.46, $P = 8.4 \times 10^{-6}$). For TEs showing strong up-regulation in *rhi* mutant testes, we observed a concurrent, pronounced loss of antisense piRNAs (e.g., *mdg3*, *invader3* and *copia*). Notably, there are antisense piRNAs against several TE families (e.g., *BS2*, *Transib2*, *invader6*) that show over 100-fold reduction in *rhi* mutant testes, a magnitude not observed for any TE family in *rhi* mutant ovaries (Fig 3I). Finally, sense piRNAs were also lost for many TE families (S5 Fig), consistent with dual-strand clusters producing piRNAs from both genomic strands in a Rhi-dependent manner. These results show that efficient production of TE-silencing piRNAs in testis depends on the RDC complex, without which many TEs are de-repressed, causing DNA damage and germ cell death in testis.

## RDC complex is required for piRNA production from dual-strand clusters in early male germ cells

To understand the role of RDC complex in piRNA cluster expression in testis, we analyzed effects of *rhi* mutation on piRNA production from major piRNA clusters. Genomic loci that generate piRNAs in testis were recently *de novo* defined leading to identification of several novel piRNA clusters [20]. Rhi was dispensable for expression of uni-strand piRNA clusters, *flam* and *20A* (Fig 4A), similar to results from ovary studies [2,4,14]. Surprisingly, we found that Rhi was also dispensable for piRNA production from the Y-linked *Su(Ste)* locus, which is the most active piRNA cluster in testis [20]. Unlike *flam* and *20A*, *Su(Ste)* is a dual-strand cluster that generates piRNAs from both genomic strands. We confirmed by RNA fluorescence *in situ* hybridization (FISH) that piRNA precursor transcription from *Su(Ste)* appeared intact in testes without Rhi (Fig 4C). In contrast to *Su(Ste)*, piRNA production from other major dual-strand clusters, including the Y-linked *petrel* cluster, collapses in *rhi* mutant testes (Fig 4A), indicating that expression of the majority of dual-strand piRNA clusters in testis relies on Rhi. Interestingly, dependence of dual-strand cluster expression on Rhi varies between sexes: *38C* is more affected by loss of *rhi* than *42AB* in testis, while the opposite is found in ovary. Furthermore, piRNA production from both strands of complex satellites, which we recently found to behave as dual-strand piRNA clusters [20], also drastically declined in *rhi* mutant testes and ovaries (Fig 4B). These results show that Rhi is essential for piRNA production from a large fraction of piRNA clusters in the male germline.

Since RDC complex forms distinct foci in the nuclei of germ cells (Fig 2A), we set out to test if Rhi binds the chromatin of dual-strand clusters whose expression depends on RDC complex, as reported in ovary [2,4,14]. Given that expression of RDC complex is restricted to a small number of cells at the apical tip of testis, we used *bam* mutant testes, where Rhi-expressing spermatogonia are expanded (Fig 2E), to perform ChIP-seq of Rhi. All major dual-strand clusters, with the exception of *Su(Ste)*, were enriched for Rhi binding (Fig 4D). In agreement with ChIP-seq, independent ChIP-qPCR showed no evidence of Rhi binding on *Su(Ste)* locus (n = 4) (Fig 4G). Rhi was also absent on chromatin of uni-strand clusters, *flam* and *20A*. Importantly, the binding of Rhi on different loci seems to correlate with its effect on promoting piRNA cluster expression. Rhi does not bind, and is dispensable for piRNA production from, uni-strand clusters and *Su(Ste)*, while it binds, and is required for expression of, other dual-strand clusters (Fig 4A and 4D). Also, dual-strand clusters that show the highest levels of overall Rhi binding, *38C* and *AT-chX*, demonstrate the strongest Rhi dependence for piRNA production. To characterize the relationship between Rhi binding and piRNA production on a genome-wide scale, we analyzed Rhi binding and piRNA production in 1Kb genomic windows spanning the entire genome (Fig 4E). For loci that depend on Rhi to produce piRNAs, we observed a strong correlation between Rhi binding and piRNA levels (Spearman's $\rho = 0.97$; Pearson's $\rho = 0.99$). On the other hand, loci that continue to produce piRNAs at the absence of

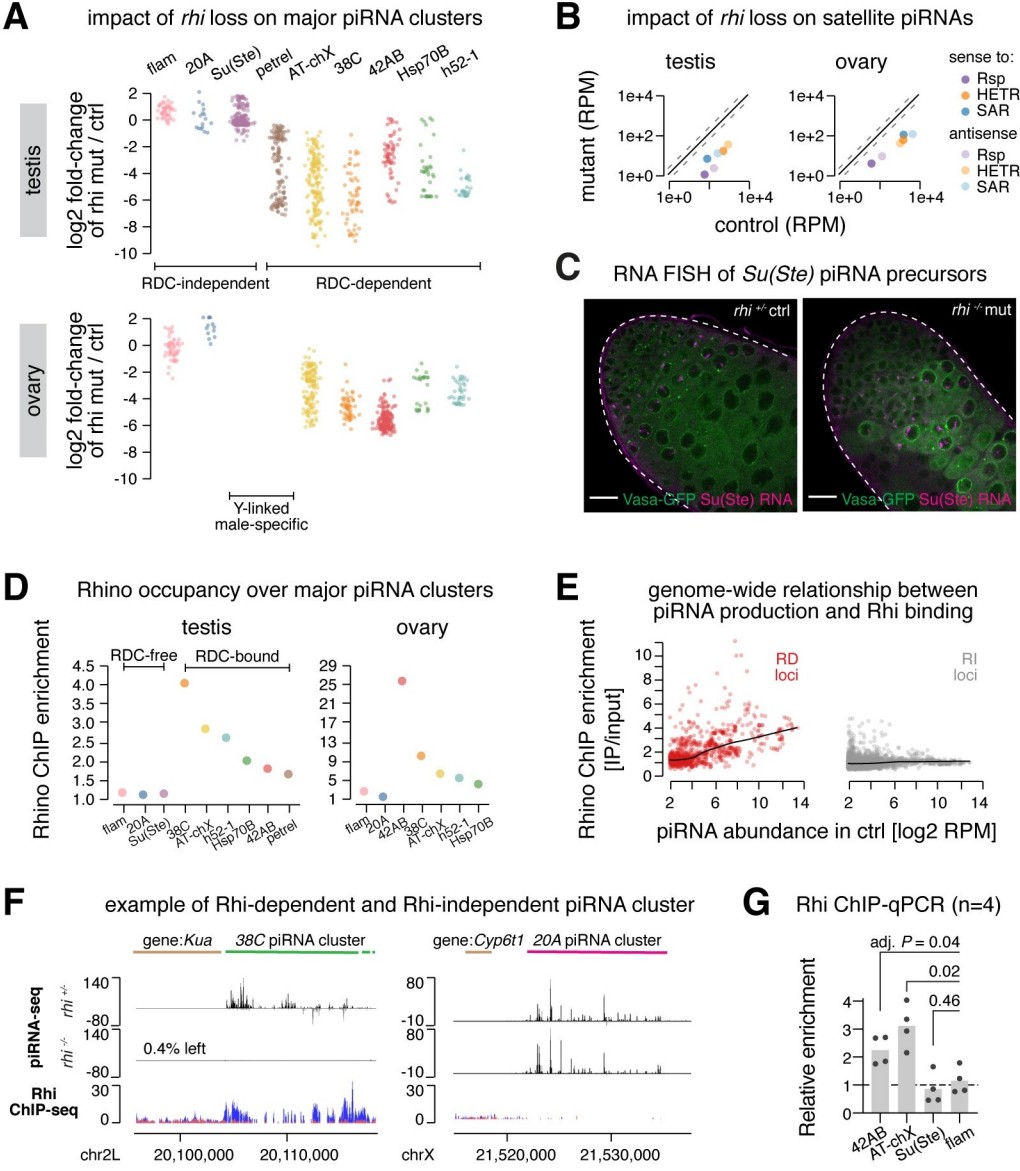

**Fig 4. RDC complex is required for piRNA production from dual-strand piRNA clusters.** (A) Impacts of *rhi* loss on piRNA production from major piRNA clusters in testis and ovary. Scatter plot showing fold-change of piRNA production from 1Kb genomic windows spanning major piRNA clusters in testis (top) and ovary (bottom), upon loss of *rhi*. Two Y-linked, male-specific clusters are not present in female genome. Each cluster is given a unique color. Averages of two biological replicates are shown. (B) Impact of *rhi* loss on piRNA production from satellite repeats in testis and ovary. Scatter plot showing levels of complex satellite-mapping piRNAs in *rhi²/KG* mutant and control testis (left) and ovary (right). Each complex satellite is assigned a color, and piRNAs sense to satellite consensus are marked with higher opacity than antisense ones. Mean of two biological replicates is shown. (C) Unperturbed expression of *Su(Ste)* piRNA precursors in testes of *rhi* mutants. RNA fluorescence *in situ* hybridization of *Su(Ste)* piRNA precursors in *rhi* control (left) and *rhi²/KG* mutant (right) testes that express Vasa-GFP. Scale bar: 20μm. (D) Rhi occupancy over major piRNA clusters in testis and ovary. Scatter plot showing Rhi ChIP-seq enrichment over major piRNA clusters in testis (left) and ovary (right). Shown are averages of two biological replicates. Two Y-linked, male-specific clusters are not present in female genome. Each cluster is colored the same way as in (A). (E) Scatter plot showing the relationship between Rhi ChIP enrichment and piRNA production over 1Kb genomic windows. Loci that show ≥4-fold decline in piRNA production at the absence of *rhi* are defined as Rhi-dependent ("RD", red), otherwise Rhi-independent ("RI", gray). Each dot is the average of two biological replicates. Black lines show local regression. (F) Examples of Rhi-dependent (*38C*, left) and Rhi-independent (*20A*, right) piRNA clusters. piRNA-seq in *rhi* mutant and control testes are shown at the top, and Rhi ChIP-seq is shown at the bottom (blue: IP, red: input). The profile of a representative replicate is shown. (G) Rhi does not bind *Su(Ste)* cluster in testis. Bar graphs showing Rhi ChIP-qPCR (n = 4) over four piRNA clusters in testis. Adjusted *P*-values are from multiple t-tests corrected for multiple comparisons by the Holm-Sidak method. Uni-strand piRNA cluster *flam* not bound by Rhi serves as a negative control.

Rhi usually have little, if any, Rhi binding. Collectively, our data indicate that Rhi physically binds the chromatin and ensures the expression of dual-strand piRNA clusters, with a notable exception of *Su(Ste)*.

## Sexually dimorphic genome occupancy of RDC complex sculpts sex-specific piRNA program

piRNA profiles are distinct in male and female gonads, and expression of dual-strand piRNA clusters are sexually dimorphic [20]. To explore if RDC complex might be involved in orchestrating sex-specific piRNA programs, we profiled Rhi binding on the genome in ovaries under identical ChIP-seq conditions as in testes. This analysis revealed differences in Rhi genome occupancy between sexes among top piRNA clusters (Fig 5A). For example, Rhi is more enriched on *38C* than *42AB* in testes, whereas the reciprocal is seen in ovaries, correlating with differential piRNA production from these two loci in two sexes. In addition, *80EF* and *40F7* clusters have high levels of Rhi binding in ovary but low in testis, mirrored by abundant piRNA production from these two loci in ovary but not in testis. Finally, an ovary-specific dual-strand piRNA cluster, *Sox102F*, is bound by Rhi in ovary, while there is no evidence of Rhi binding at *Sox102F* in testis where it is inactive (Fig 5B). Altogether, the observed link between Rhi binding and piRNA production between males and females suggests that the sex-specific Rhi binding on piRNA clusters is responsible for sculpting a sexually dimorphic piRNA program.

## RDC complex enables dynamic piRNA production during spermatogenesis

ChIP-seq provides the genome-wide profile of Rhi binding, but it masks possible differences of Rhi localization among individual cells. Imaging of Rhi revealed distinct Rhi localization in nuclei of germ cells at different stages of spermatogenesis (Fig 6A). In the nuclei of GSCs and

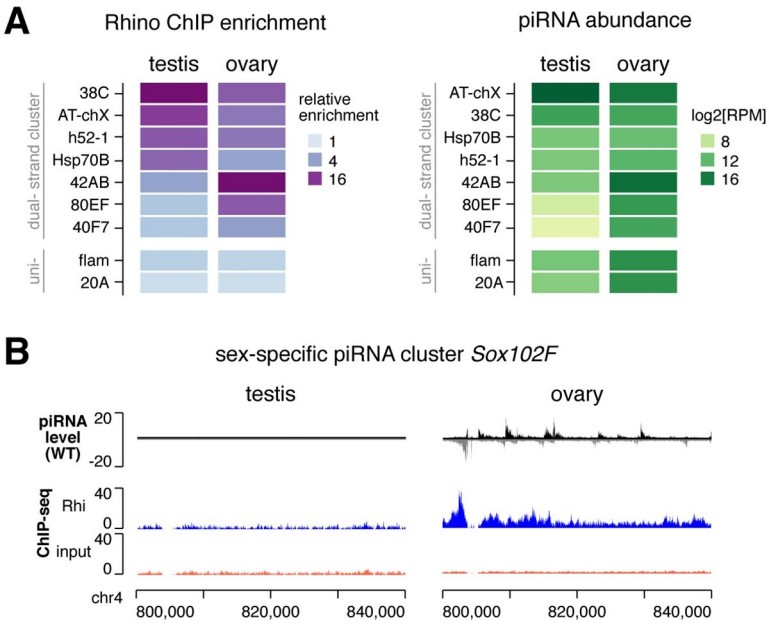

**Fig 5. Sexually dimorphic RDC genome occupancy sculpts sex-specific piRNA program.** (A) Heatmaps showing relative enrichment of Rhi binding over major piRNA clusters determined by ChIP-seq (left) and piRNA production from major piRNA clusters (right) in two sexes. The mean of two biological replicates is shown. Rhi-dependent dual-strand clusters are shown at the top, and Rhi-independent uni-strand clusters at the bottom. (B) *Sox102F* piRNA cluster produces piRNAs exclusively in ovary. Rhi is enriched on *Sox102F* in ovary, but not in testis. Shown is the profile from a representative replicate.

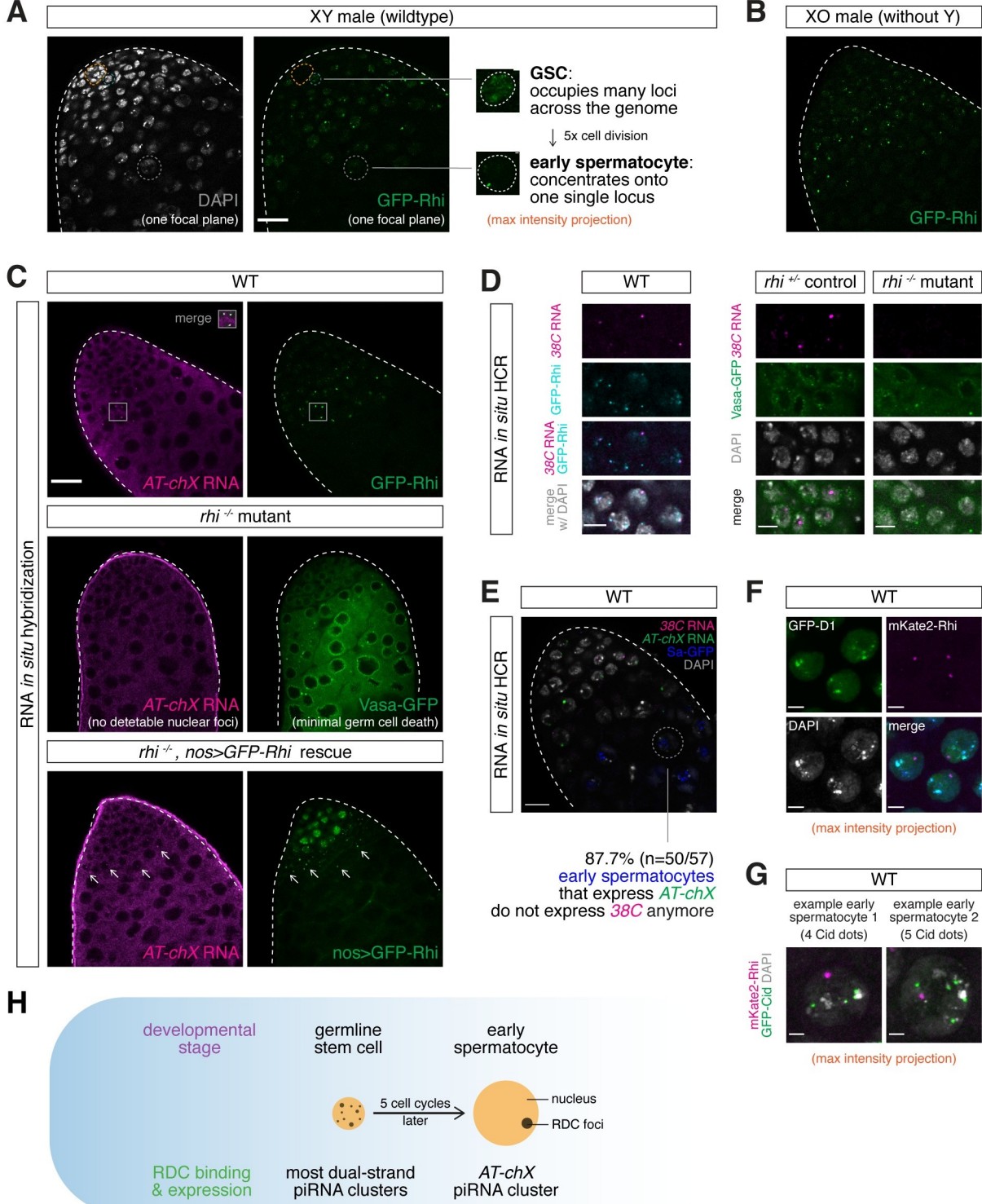

**Fig 6. RDC complex enables dynamic piRNA production during early spermatogenesis.** (A) Rhi localizes to multiple nuclear foci in GSCs and early spermatogonia but concentrates in a single dot in early spermatocytes. Confocal images of the apical tip of testis expressing GFP-Rhi transgene. A single focal plane is shown for DAPI and GFP-Rhi, while maximum intensity projections covering entire nuclei are shown for a GSC and an early spermatocyte on the right. Orange circle outlines the somatic hub for identification of GSCs next to it. Early spermatocyte is identified by formation of chromosome territory and increased nuclear size by DAPI. (B) Rhi localization to a single nuclear dot is not affected in early spermatocytes that lack Y chromosome. Confocal image of the apical tip of testis from XO male (that lacks Y chromosome) expressing

GFP-Rhi transgene. (C) Rhi localizes exclusively to *AT-chX* piRNA cluster in early spermatocytes. Top: RNA fluorescence *in situ* hybridization (FISH) of *AT-chX* piRNA precursors in wildtype testis expressing GFP-Rhi. Middle: RNA FISH of *AT-chX* piRNA precursors in *rhi²/KG* mutant testis. Note that the absence of *AT-chX* RNA foci is not due to the loss of germ cell as Vasa expression shows minimal germ cell death in this particular testis. Bottom: expression of Rhi transgene driven by *nos-Gal4* rescues expression of *AT-chX* cluster in testis of *rhi²/KG* mutant. (D) Rhi binds *38C* piRNA cluster in spermatogonia. RNA *in situ* hybridization chain reaction (HCR) of *38C* piRNA precursors in testis expressing GFP-Rhi (left) or Vasa-GFP in heterozygous control and *rhi* mutant background (right). Shown are spermatogonia as indicated by DAPI staining. Note that, in contrast to exclusive co-localization with *AT-chX* cluster in spermatocytes shown in (C), only a subset of Rhi foci co-localize with *38C* RNA foci, indicating that Rhi binds other piRNA clusters besides *38C* in spermatogonia. *38C* RNA signal is present in control but absent in *rhi* mutant testes. (E) *AT-chX* and *38C* co-express in spermatogonia, but only *AT-chX* is expressed in most early spermatocytes. Dual *in situ* HCR of *38C* and *AT-chX* piRNA precursors in testis expressing Sa-GFP (marker for spermatocytes). Circled is an example of early spermatocytes that express *AT-chX* but not *38C*. Quantification is shown at the bottom. Note that co-expression of *38C* and *AT-chX* can be seen in Sa-negative spermatogonia. (F) D1 and Rhi do not co-localize in early spermatocytes. Confocal image of early spermatocytes expressing GFP-D1 (a protein trap line) and mKate2-Rhi driven by the *rhi* promoter. (G) Rhi is localized in a single dot in spermatocyte nuclei, where individual chromosomes occupy distinct chromosomal territories. Confocal image of early spermatocytes expressing mKate2-Rhi and the centromere marker GFP-Cid. The presence of four or more Cid foci indicates that centromeres of individual non-homologous chromosomes are not clustered together. (H) Proposed model of how Rhi switches genomic binding sites during gametogenesis from GSC to early spermatocyte stage, in order to allow dynamic employment of different piRNA clusters. Scale bars: 20μm (A-C, E), 5μm (D), 4μm (F) and 2μm (G).

spermatogonia, Rhi forms many discrete foci, suggesting–in agreement with ChIP-seq results–that it binds multiple genomic loci. As male germ cells differentiate into spermatocytes and prepare for meiosis, however, Rhi concentrates as one single dot in nuclei of early spermatocytes, suggesting its specific localization at one single locus. Notably, while homologous chromosomes pair in about half of late spermatogonia and very early spermatocytes [21], we only observed one bright Rhi dot in virtually all nuclei at these stages, suggesting that this locus resides on one of the two sex chromosomes rather than an autosome. To explore where Rhi binds at this stage, we first examined testes from XO males that lack the Y chromosome and found that Rhi still localizes as one single dot in early spermatocytes, arguing against Y-linked loci such as *Su(Ste)* and *petrel* (Fig 6B). Simultaneous imaging of Rhi and RNA FISH of transcripts from the X-linked *AT-chX* locus revealed co-localization of *AT-chX* nascent transcripts and Rhi in one single dot from late spermatogonia to early spermatocytes, indicating that Rhi concentrates on *AT-chX* locus at this stage (Fig 6C). Indeed, single *AT-chX* RNA foci in individual nuclei became undetectable in *rhi* mutant testes, and expressing GFP-tagged Rhi transgene by *nanos-Gal4* in *rhi* mutant background restored *AT-chX* expression (Fig 6C). Importantly, even though *nanos-Gal4* drives stronger expression of GFP-Rhi in earlier stages (GSC and early spermatogonia), *AT-chX* transcripts remained highly expressed specifically during later stages (late spermatogonia and early spermatocyte) (Fig 6C). This finding suggests that the spatio-temporally regulated gene expression of *AT-chX* piRNA cluster is rather robust to perturbations to the Rhi protein level, and the low expression of *AT-chX* piRNA cluster earlier in GSCs and early spermatogonia is not limited by the level of Rhi protein. In sum, these results show that Rhi binds multiple genomic loci in GSCs and spermatogonia but appears to concentrate on a single *AT-chX* locus later.

As Rhi is required for non-canonical transcription of dual-strand piRNA clusters, depletion of Rhi from clusters other than *AT-chX* should cease their expression. To test this, we set out to conduct RNA FISH of piRNA precursors from other Rhi-dependent dual-strand piRNA clusters (Fig 5A). Previously, FISH detection of piRNA precursor transcripts was performed in polyploid nurse cells in fly ovary, whose genome is endo-replicated up to 1032C with a much higher expression level of piRNA precursor transcripts [2,8,22]. In addition to low expression levels in the diploid male germline, piRNA cluster transcripts are difficult to target, as they are highly repetitive and share extensive sequence homology with TEs. To tackle these challenges, we employed *in situ* hybridization chain reaction (HCR), which permits enzyme-free signal amplification and automatic background suppression [23], to target transcripts from *38C*

piRNA cluster that has a relatively high expression level in testis (Fig 5A). *in situ* HCR successfully detected nascent *38C* piRNA precursors that co-localized with Rhi in diploid male germ cells (Fig 6D). These signals from nascent *38C* piRNA cluster transcripts were absent in *rhi* mutant testes, confirming the specificity of our probes (Fig 6D). We found that both *38C* and *AT-chX* piRNA precursors can be seen in nuclei of spermatogonia. In contrast, only *AT-chX*, but not *38C*, cluster continues to be expressed in 87.7% (n = 50/57) of early spermatocyte nuclei (Fig 6E). Therefore, the expression of *38C* piRNA cluster is turned off when Rhi concentrates onto the *AT-chX* locus.

During early spermatogenesis, satellite DNAs located at peri-centromeric heterochromatin of different chromosomes cluster together to form distinct nuclear compartments called chromocenters [24,25]. To explore the possibility that the repetitive *AT-chX* locus is also recruited to a chromocenter, we compared Rhi localization with that of D1, which binds the AATAT satellite on chromosome X, Y and 4 [24]. We found that D1 does not co-localize with the single Rhi dot in early spermatocytes (Fig 6F). We also analyzed localization of Cid, a centromere-specific histone H3 variant that functions as an epigenetic mark for the centromere identity [26]. The single Rhi dot is present in nuclei that contain four or more separate Cid foci (Fig 6G), indicating that non-homologous chromosomes are separated and occupy distinct chromosome territories in these nuclei. These results show that the *AT-chX* locus bound by Rhi occupies a nuclear compartment distinct from chromocenters.

Taken together, our results suggest that RDC complex binds, and thus enables, the expression of many dual-strand piRNA clusters from GSCs to spermatogonia, including *38C*, but gradually concentrates onto a single locus, *AT-chX*, in early spermatocytes (Fig 6H). Dynamic association of RDC with piRNA clusters during early spermatogenesis executes a fluid piRNA program to allow different piRNA clusters to be engaged at different developmental stages.

## Discussion

Most proteins involved in piRNA pathway in *Drosophila* were initially identified in screens for female sterility or TE de-repression in ovaries [9,10,27–30]. However, the first described case of piRNA repression, silencing of *Stellate* by *Su(Ste)* piRNAs during spermatogenesis, indicates that piRNA pathway operates in gonads of both sexes [31,32]. Many proteins involved in piRNA pathway in ovaries are also required for male fertility and *Stellate* silencing in testes, supporting the conservation of piRNA pathway machinery between sexes [31,33–39]. Notably, a few proteins stood out as exceptions: Rhi, Del and Cuff that form a complex to enable transcription of dual-strand piRNA clusters in ovary [2,4,14], for which no fertility defects or *Stellate* de-repression were observed in mutant males [10,13,14]. This suggested that molecular mechanisms controlling piRNA cluster expression in testis might be different from ovary. Our results, however, demonstrate that RDC complex is assembled in testis (Fig 2) and required for TE silencing (Fig 3) in male germline, indicating that the molecular machinery regulating piRNA cluster expression is conserved between sexes.

### piRNA pathway in *Drosophila* testis: beyond *Stellate* silencing

Genomic loci that encode ovarian piRNAs were systematically identified across the genome in several studies [2,40]. In contrast, piRNA studies on testes were mostly focused on a single locus, *Su(Ste)*, that encodes piRNAs to silence *Stellate*, and no systematic search for piRNA clusters in testis has been performed to date. We recently *de novo* identified piRNA clusters in testis [20], laying the foundation for broader understanding of piRNA biogenesis and function in male gonads. We found that RDC is essential for expression of all major dual-strand piRNA clusters in testis, with a remarkable exception of *Su(Ste)* (Fig 4). This explains the previous

observation that *rhi* and *cuff* are dispensable for *Stellate* silencing [14,15]. In contrast, many other piRNA pathway factors such as Aub, Ago3 and Zuc are involved in both *Stellate* and TE repression [31,33,35]. Mutations in these genes cause dramatic disruption of spermatogenesis, often leading to complete male sterility. In comparison, *rhi*, *del* and *cuff* mutant males demonstrate milder fertility defects, suggesting that *Stellate* de-repression is the major cause of spermatogenesis failure in other piRNA pathway mutants. Indeed, previous studies demonstrated that *Stellate* de-repression induced by deleting the *Su(Ste)* locus alone, without global perturbations of piRNA pathway, disrupts spermatogenesis and causes male sterility [41]. Thus, the finding that RDC is dispensable for *Stellate* repression provides a unique opportunity to understand impacts of silencing other piRNA targets in the male germline.

Our results indicate that piRNA-guided repression plays a crucial role in spermatogenesis beyond *Stellate* silencing, as *rhi*, *del* and *cuff* mutant males show rapid fertility decline, germline DNA damage and severe loss of germline content including GSCs (Fig 3). These phenotypes are likely caused by TE de-repression and the resultant genome instability, though we also identified complex satellites and a host protein-coding gene, SUMO protease *CG12717/pirate*, as targets of piRNA silencing in testes [20]. In the future, it will be important to disentangle the contributions of de-repressing different piRNA targets to spermatogenesis defects observed in testes of piRNA pathway mutants.

Even though the fertility of *rhi* mutant males is substantially compromised, they produce a small number of functional spermatozoa, at least when they are young (Fig 1). In contrast, females lacking *rhi*, *del* or *cuff* are completely sterile [10,13]. This distinction between the two sexes might result from differences in the TE threat faced by male and female germ cells. Different TE families are activated upon disruption of piRNA pathway in ovary and testis and, generally, there is a stronger TE threat in ovary [20]. Differential TE de-repression in two sexes might be responsible for stronger defects in oogenesis at the absence of RDC. Alternatively, it might reflect differences in DNA damage response in two sexes. An egg is energetically more expensive to make than a spermatozoon. In line with this, DNA damage responses (activated by e.g., TE transposition) often arrest oogenesis to avoid wasting resources on a defective egg [42,43]. Since oogenesis is usually shut down to attempt repair, incomplete oogenesis results in female sterility. In contrast, quality control mechanisms of spermatogenesis frequently kill unqualified germ cells [44], without pausing the developmental program of those surviving ones. Because spermatogenesis permits a large number of germ cells to develop in parallel, even though the unqualified ones are killed, a few surviving germ cells might be able to complete sperm development. Together, differential TE threats coupled with distinct response strategies to DNA damage might underlie sex-specific sterility when RDC is lost.

## *Su(Ste)*: an RDC-independent dual-strand piRNA cluster free of RDC binding

*Su(Ste)* locus on Y chromosome is the most prolific source of piRNAs in testes [20]. piRNAs are generated from both genomic strands of *Su(Ste)* repeats, making it akin to other dual-strand piRNA clusters [31,45]. However, our results showed that RDC is dispensable for expression of *Su(Ste)* piRNAs, while it is required for piRNA production from all other dual-strand clusters in testis (Fig 4A) and ovary [2,4]. What might explain the ability of *Su(Ste)* to generate piRNAs in an RDC-independent fashion? RDC ensures transcription of piRNA precursors by suppressing their premature termination [3] and promoting non-canonical transcription initiation [8]. Interestingly, the structure of *Su(Ste)* locus is different from other dual-strand piRNA clusters, as it is composed of many almost identical, relatively short units, all of which are flanked by two canonical, albeit convergent, promoters driving expression of both

genomic strands [31]. In fact, sense transcripts of *Su(Ste)* were found to be spliced and polya-denylated [31], consistent with the absence of RDC, which suppresses splicing and polyadenylation [2–4], at this locus. Thus, the presence of canonical promoters flanking individual short units of *Su(Ste)* repeats might enable their expression without engaging RDC complex.

Consistent with a role in promoting piRNA precursor transcription, Rhi is enriched on chromatin of dual-strand piRNA clusters. Such a correlation between direct Rhi binding and Rhi-dependent piRNA expression was previously reported in ovary [2]. The function of RDC explains why piRNA production depends on its presence on chromatin; however, the molecular mechanism responsible for specific recruitment of RDC to piRNA clusters remained poorly understood. It has been shown that piRNAs expressed during oogenesis are deposited into the oocyte and play an important role in jump-starting piRNA biogenesis in the progeny [46,47]. Maternally supplied piRNAs were shown to induce deposition of Rhi on cognate genomic locus in the progeny [7]. Thus, piRNAs expressed from the cluster and RDC binding to the cluster seem to form a positive feedback loop: RDC is required for piRNA production, and piRNAs in turn guide deposition of Rhi on cognate genomic loci. piRNA-dependent Rhi deposition might be mediated by the nuclear Piwi protein that directs the establishment of histone H3K9me3 mark [48–51], which provides a binding site for Rhi chromo-domain [2,7]. Importantly, Piwi- and piRNA-dependent Rhi recruitment seems to occur in a narrow developmental window during early embryogenesis. This was demonstrated by the observation that depleting Piwi during early embryogenesis is sufficient to perturb Rhi localization on piRNA clusters, while depleting Piwi during larval or adult stages does not change Rhi localization [52]. Our finding that Rhi is localized to all dual-strand piRNA clusters in testis except *Su(Ste)* (Fig 4D) is compatible with an idea that Rhi binding to genomic loci in the zygote is guided by maternal piRNAs. Indeed, in contrast to most other dual-strand clusters that are active in the germline of both sexes, Y-linked *Su(Ste)* locus generates piRNAs only in males. Therefore, unlike other piRNAs, *Su(Ste)* piRNAs are not deposited into the oocyte, resulting in the inability to recruit Rhi to *Su(Ste)* in the progeny. It is interesting to compare *Su(Ste)* with another dual-strand piRNA cluster on Y chromosome, *petrel*. Unlike *Su(Ste)*, Rhi is enriched on *petrel* chromatin and piRNA production from this cluster depends on Rhi (Fig 4A and 4D). However, *petrel* piRNAs derived from male-specific Y should be absent in ovaries and hence no *petrel* piRNAs can be deposited into the oocyte. At the first glance, these observations argue against a possibility that maternal piRNAs guide Rhi deposition on *petrel* cluster. However, unlike *Su(Ste)*, *petrel* is enriched of different TE sequences. As a result, TE-mapping piRNAs produced from other clusters in ovary might be able to target *petrel*. Indeed, piRNAs mapping to TEs present at the *petrel* locus can be found in unfertilized eggs [46]. For example, *roo* piRNAs are the most abundant TE-mapping, maternally deposited piRNAs in the early embryo [46]. Several *roo* fragments are present at *petrel*, and these sequences occupy ~6% of the total length of this cluster, making *roo* the third most represented TE at *petrel* (after *IDEFIX* and *ninja*) [20]. piRNAs against other TEs located at *petrel* are also deposited in the embryo. Overall, while our results did not directly address the mechanism of Rhi recruitment to specific genomic loci, they show that studying Rhi occupancy on Y-linked piRNA clusters provides a novel angle to study this problem.

## Dynamic organization of piRNA pathway during spermatogenesis

Our results showed that components of RDC complex are expressed exclusively in male germline during early stages of spermatogenesis, from GSCs to early spermatocytes (Figs 1 and 2). Interestingly, expression of Piwi and Ago3, two of the three PIWI proteins in *Drosophila*, is also restricted to the same developmental stages [34,53]. Piwi is required for piRNA-guided

transcriptional silencing in the nucleus [48–51], while Ago3 is involved in heterotypic ping-pong cycle in cytoplasm [33,40,54], indicating that these processes operate in the same cells that have RDC-dependent transcription of piRNA clusters. In contrast to RDC, Piwi and Ago3, expression of the third PIWI protein, Aub, continues through spermatocyte stage until meiosis [34]. How can developing male germ cells be protected when the piRNA pathway is greatly simplified? It is possible that the silencing network initiated by piRNA pathway factors earlier can self-sustain later. piRNAs produced with the help of RDC complex and Ago3-dependent heterotypic ping-pong could persist and continue to function through spermatocyte differentiation, as long as they load onto Aub.

Interestingly, the cessation of RDC, Piwi and Ago3 expression during spermatogenesis coincides with the mitosis-to-differentiation transition. This transition is accompanied by one of the most dramatic changes in gene expression programs, with the general transcriptional machinery replaced by the testis-specific ones like tTAF and tMAC [55,56]. Thus, the following stage of male germline development can also be seen as a less-protected window of spermatogenesis, providing an opportunity for TEs and other selfish genetic elements to thrive.

Restriction of the piRNA pathway to early stages of spermatogenesis contrasts with its activity during oogenesis. piRNA pathway factors appear to be expressed during all stages of oogenesis from GSCs to late-stage nurse cells. Recent studies reported several new factors involved in piRNA pathway in ovary, such as Moon, Boot, Nxf3, Panx, Arx and Nxf2 [8,22,30,57–65]. These proteins are expressed at a low level in testis and their functions during spermatogenesis have not yet been reported. Our results suggest that, similar to RDC, these proteins might function in piRNA pathway in testis, and their low expression could be explained by restricted expression in early male germline. Indeed, we found that Moon co-expresses with, and acts genetically downstream of, RDC in testes (Figs 2 and S4). Overall, our results suggest that piRNA pathway machinery is likely conserved between sexes. However, the developmental organization of piRNA pathway during gametogenesis is different: whereas the entire pathway is active throughout oogenesis, processes that require RDC, Piwi or Ago3 likely terminate when mitotic spermatogonia differentiate into spermatocytes and prepare for meiosis.

## Dynamic expression of piRNA clusters during spermatogenesis

Our results revealed dynamic association of RDC complex with different genomic loci during spermatogenesis (Fig 6). In GSC and spermatogonia nuclei, RDC complex localizes to many foci, and ChIP-seq data indicate that Rhi associates with multiple dual-strand clusters. As germ cells differentiate into early spermatocytes, however, RDC gradually concentrates onto a single locus, *AT-chX*, on X chromosome. Since transcription of dual-strand piRNA clusters is dependent on RDC complex, the dynamic localization of RDC suggests that expression of piRNA clusters changes as germ cells progress from GSCs to early spermatocytes. Indeed, detection of nascent cluster transcripts revealed that *38C* is active early, but not later when most RDC concentrates onto *AT-chX*. Notably, though not dependent on Rhi, transcription of *Su(Ste)* piRNA cluster was shown to span a narrow window from late spermatogonia to early spermatocyte as well [45], likely reflecting the promoter activity that drives *Su(Ste)* transcription [31]. Dynamic expression of piRNA clusters during spermatogenesis is also supported by the study that showed spermatogonia and spermatocytes have distinct piRNA populations [66]. In contrast to dynamic localization of RDC complex and cluster expression in testes, previous work depicted a static view of piRNA production in female gonads. There has been no evidence of dynamic localization of RDC complex to different clusters during oogenesis, and all clusters appear active throughout female germline development. It will be interesting to explore whether expression of piRNA clusters changed dynamically during oogenesis.

Through studying a protein complex thought to be absent during spermatogenesis, we uncovered the sexually dimorphic and dynamic behaviors of a molecular machinery that drives dual-strand piRNA cluster expression during *Drosophila* gametogenesis.

## Materials and methods

### Fly stocks

The following stocks were used: *bam^(Δ86)* (BDSC5427), *bam^(Df)* (BDSC27403), C(1)RM (BDSC9460), *nos-Gal4* (BDSC4937), *UASp-shRhi* (BDSC35171), *iso-1* (BDSC2057), *GFP-Cid* (BDSC25047), *GFP-D1* (BDSC50850) were obtained from Bloomington Drosophila Stock Center; *GFP-Rhi* (VDRC313340), *GFP-Del* (VDRC313271), *GFP-Cuff* (VDRC313269), *moon^(Δ1)* (VDRC313735), *moon^(Δ28)* (VDRC313738), *Bam-GFP* (VDRC318001), *w^(1118)* (VDRC60000) were obtained from Vienna Drosophila Resource Center; *tj-Gal4* (DGRC104055) was obtained from Kyoto Stock Center; *rhi^2*, *rhi^(KG)* and *UASp-GFP-Rhi* were gifts of William Theurkauf; *del^(WK36)*, *del^(HN56)*, *cuff^(QQ37)* and *cuff^(WM25)* were gifts of Trudi Schüpbach; *GFP-Vasa* (gift of Paul Lasko), *copiaLTR-lacZ* (gift of Elena Pasyukova), *GFP-Moon* (gift of Peter Andersen), *Sa-GFP* (gift of Xin Chen). *UASp-mKate2* was described before [3]. GFP-tagged Rhi, Del, Cuff and Moon are previously described transgenes constructed by inserting N-terminal GFP into genomic BACs via recombineering [2,8]. XO males were generated by crossing *GFP-Rhi* males to C(1)RM females. To perform GFP-Rhi ChIP, *bam^(Δ86)* and *GFP-Rhi* were recombined.

### Generation of transgenic flies

To make mKate2-tagged Rhi driven by endogenous *rhi* promoter, the ~2Kb region upstream of *rhi* gene that includes the putative endogenous *rhi* promoter [4] was cloned from genomic DNA of *Drosophila melanogaster* by PCR (forward primer: AGG CCT ATG TAC CAA GTT GTT AAC TCT ATC G, reverse primer: GGT ACC AGA CAT AAC TTA TCC GCT CAC AGG). PCR product was digested by Stu1 and Kpn1, then ligated into Stu1- and Kpn1-digested vector that contains mKate2-Rhino, mini-white gene and the ΦC31 attB site. The construct was inserted into genomic site 76A2 (y1 w1118; PBac{y+-attP-9A}VK00013) on chromosome 3 using ΦC31-mediated recombination.

### Sperm exhaustion test

The test was modified from Sun et al. (2004) and done at 25˚C. Individual 1-day old virgin males (either *rhi^(2/KG)* or heterozygous siblings, n = 15) were allowed to mate with three 4-day old wildtype virgin females (*iso-1*) for 24hrs. Each male was then moved to mate with another three 4-day old wildtype virgin females (*iso-1*) for another 24hrs, and this was repeated every 24hrs for a total of 14 days. Since each male encountered multiple young virgin females every day, their sperm were exhausted, and the number of progeny produced by females can be used to represent the daily male fertility. Inseminated females were flipped every other day and kept for 20 days (without contact with other males) to achieve maximal egg laying. A binary result of whether there is offspring or not was used to approximate whether a male is fertile or not on a given day, and we plotted the percentage of fertile male out of 15 tested males each day for *rhi^(2/KG)* and heterozygous siblings. To probe the male fertility more quantitatively, we repeated the test with five males for ten days. Instead of recording a binary result, we counted the number of progeny. The number of adult offspring was counted 15 days after female fly removal to allow most laid eggs to develop into adulthood. The averaged total number of offspring for each male each day was plotted for each genotype. This was repeated for *del^(HN/WK)*, *cuff^(WM25/)*

$^{QQ37}$ and respective heterozygous controls (n = 5 per genotype), where each 1-day-old virgin male was mated with two 4-day-old $w^{1118}$ virgin females every day for a total of eleven days. After male removal, inseminated females were flipped every 3 days for a total of 15 days for counting.

## Female fertility test

Each 1-day-old virgin female with the genotype of interest was allowed to mate with two 5-day-old $w^{1118}$ males for four days. Next, flies were discarded but vials were kept for another 14 days before counting, so eggs laid had 14–18 days to develop to adulthood. The number of adult flies from each vial was counted to approximate the female fertility. For each of the four groups shown in S2 Fig, all three genotypes (control, mutant and rescue) were siblings from the same cross with similar genetic backgrounds. Results were obtained from three biological replicates (n = 3).

## Immunofluorescence staining

Testes were dissected in PBS, fixed in 4% formaldehyde for 20mins and washed by PBSTw (PBS with 0.1% Tween-20) for 3 times. Permeabilization of testes was done by incubation with PBST (PBS with 0.5% Triton-X) for 30mins. Testes were then blocked by 5% BSA in PBSTw for at least an hour, before incubation with primary antibody in 5% BSA in PBSTw at 4˚C overnight. Testes were washed 3 times with PBSTw and incubated with secondary antibody in 5% BSA in PBSTw at room temperature for 2hrs, followed by another 3 washes with PBSTw. Before mounting in VECTA-SHIELD, testes were stained by DAPI (1:5000) for 10mins and rinsed once with PBS. The following primary antibodies were used: mouse anti-Fas3 (7G10, 1:200), mouse anti-γH2Av (UNC93-5.2.1, 1:400) and rat anti-Vasa (concentrated, 1:100) were obtained from Developmental Studies Hybridoma Bank.

## RNA fluorescence *in situ* hybridization (RNA FISH)

RNA FISH was done as described previously [67]. Fixed testes were prepared as above for immunofluorescence staining. Permeabilization was done by incubation in PBST with 0.5% sodium deoxycholate for an hour, followed by 3 washes of PBSTw. Testes were transferred to first 25% and then 50% formamide, both for 10mins. Next, testes were prehybridized in hybridization buffer (50% formamide, 0.5mg/ml yeast tRNA, 0.2mg/ml heparin) for 1hr at 42˚C, before incubation with 0.1–1μg DIG-labeled RNA probe in 50μl hybridization buffer overnight at 42˚C with shaking. Testes were rinsed twice with 50% formamide for 20mins at 42˚C and then transferred to wash in PBSTw for 4 times. Subsequent blocking, staining by sheep anti-DIG antibody (PA1-85378, 1:200, Life Technologies) and mounting were the same as described above for immunofluorescence staining. DIG-labeled RNA probes were transcribed by T7 according to manufacturer's instructions. DNA template was made from genomic PCR using primers listed below, with T7 promoter sequence added 5' to the reverse primers.

   *Su(Ste)* [45]

F: 5'-CAGGTGATTACCACTATTAACGAAAAGTATGC

R: 5'-ATCCTCGGCCAGCTAGTCCT

   *AT-chX* [67]

F: 5'-AGCGATCCCACTGCTAAAGA

R: 5'-ATAAAAGGTGACCG-GCAACG

### RNA *in situ* hybridization chain reaction (HCR)

A kit containing a DNA probe set, a DNA probe amplifier and hybridization, amplification and wash buffers were purchased from Molecular Instruments (molecularinstruments.org) for *AT-chX* and *38C* transcripts. To minimize off-targets, we designed probes targeting unique regions at *AT-chX* and *38C*. For *38C*, we specifically targeted junction sites of two different TEs, the simultaneous presence of which is required to generate amplified HCR signals. The *AT-chX* (unique identifier: 3893/E038) and *38C* (unique identifier: 4026/E138-E140) probe sets initiated B1 (Alexa 647) and B3 (Alexa 546) amplifiers, respectively. *In situ* HCR v3.0 [23] was performed according to manufacturer's recommendations for generic samples in solution.

### X-gal staining

Testes were dissected in PBS, fixed in 0.5% glutaraldehyde containing 1mM $MgCl_2$ for 5mins and washed twice in PBS. Testes were incubated with 0.02% X-gal in X-gal buffer (1mM $MgCl_2$, 150mM NaCl, 10mM $Na_2HPO_4$, 10mM $NaH_2PO_4$, 3.5mM $K_4Fe(CN)_6$ and 3.5mM $K_3Fe(CN)_6$) at 37°C in dark for time of interest. Staining of *copiaLTR-lacZ* in *rhi*[2/KG], *del*[HN/WK] and *cuff*[WM25/QQ37] typically took 1.5–2.5hrs to develop. Reaction was then stopped by two washes of PBS and mounted as above for RNA FISH.

### Image acquisition and analysis

Images were acquired using confocal microscope Zeiss LSM 800 with 63x oil immersion objective (NA = 1.4) and processed using the software Fiji [68]. X-gal stained testes were imaged with 10x objective (NA = 0.3). Maximum-intensity z-projection was done in Fiji, and line intensity profiles were obtained in Fiji. All images shown were from single focal planes, unless otherwise stated. Dotted outlines were drawn for illustration purposes. To quantify the number of GSCs, we stained the hub by Fas3. Z-stacks were acquired with 0.5μm intervals to cover depths well above and below the entire hub. Vasa-positive germ cells directly adjacent to the hub in 3D were deemed as GSCs and manually counted for each testis. Even though a molecular marker for GSCs was not used, any bias in GSC counting should be shared by both *rhi*[2/KG] and heterozygous sibling controls.

### RNA-seq and analysis

RNA was extracted from dissected testes of 0–3 days old *rhi*[2/KG] and heterozygous sibling controls using TRIzol (Invitrogen). About 1μg RNA for each sample was subject to polyA+ selection using NEBNext Poly(A) mRNA Magnetic Isolation Module (NEB E7490) and then strand-specific library prep using NEBNext Ultra Directional RNA Library Prep Kit for Illumina (NEB E7760) according to manufacturer's instructions. For each genotype, two biological replicates were sequenced on Illumina HiSeq 2500 yielding 25–33 million 50bp single-end reads. Reads mapped to *D. mel* rRNA were discarded by bowtie 1.2.2 allowing 3 mismatches (<2% across all polyA-selected samples). For TE analysis, rRNA-depleted reads were mapped to TE consensus from RepBase17.08 using bowtie 1.2.2 with -v 3 -k 1. Mapped reads were normalized to the total number of reads that can be mapped to dm6 genome. Consistency between biological replicates was confirmed by >0.98 correlation coefficient, so the mean of them was used for all analyses. For simplicity, reads mapped to LTR and internal sequence were merged for each LTR TE given their well correlated behaviors. Note that polyA-selection was done for TE quantification in order to exclude piRNA precursor transcripts from dual-strand clusters and non-canonical transcripts from individual TEs, which are not polyadenylated but share sequence homology with TEs.

## piRNA-seq and analysis

RNA extraction was done as above for RNA-seq. 19-30nt small RNAs were purified by PAGE (15% polyacrylamide gel) from ~1μg total RNA. Purified small RNA was subject to library prep using NEBNext Multiplex Small RNA Sample Prep Set for Illumina (NEB E7330) according to manufacturer's instructions. Adaptor-ligated, reverse-transcribed, PCR-amplified samples were purified again by PAGE (6% polyacrylamide gel). Two biological replicates per genotype were sequenced on Illumina HiSeq 2500 yielding 15–20 million 50bp single-end reads. Adaptors were trimmed with cutadapt 2.5 and size-selected for 23-29nt sequences for piRNA analysis. 23-29nt reads that mapped to rRNA were discarded by bowtie 1.2.2 tolerating 3 mismatches (<30% in control samples). For TE analysis, 23-29nt small RNA reads were mapped and normalized as done for polyA+ RNA described above, with correlation coefficients between replicates all >0.94. Averages of two well-correlated replicates were used for all analyses. Complex satellite-mapping small RNAs were analyzed similarly (with ovary data downloaded from GSE126578). For piRNA cluster analysis, we used piRNA clusters defined in [20]. 1Kb genomic windows in individual piRNA clusters were generated with bedtools v2.28.0, and the ones including highly expressed miRNA, snRNA, snoRNA, hpRNA or 7SL SRP RNA were excluded. Coverage over individual piRNA clusters were computed using the pipeline tolerating local repeats described in [20]. A pseudo-count of 1 was added before calculating log2 fold-change of *rhi* mutant over control.

## ChIP-qPCR, ChIP-seq and analysis

ChIP protocol was modified based on Le Thomas et al. (2014). For each biological replicate, 200 pairs of 0–2 days old testes or 100 pairs of 4–5 days old ovaries (yeast-fed for 3 days) were fixed in 1% formaldehyde for 10mins, quenched by 25mM glycine for 5mins and washed 3 times with PBS. Fixed testes were snap-frozen in liquid nitrogen and stored at -80˚C before ChIP. Frozen testes were first resuspended in PBS and then washed in Farnham Buffer (5mM HEPES pH8.0, 85mM KCl, 0.5% NP-40, protease inhibitor, 10mM NaF, 0.2mM $Na_3VO_4$) twice. Testes were then homogenized in RIPA Buffer (20mM Tris pH7.4, 150mM NaCl, 1% NP-40, 0.5% sodium deoxycholate, 0.1% SDS, protease inhibitor, 10mM NaF, 0.2mM $Na_3VO_4$) using a glass douncer and a tight pestle. Sonication was done in Bioruptor (Diagenode) on high power for 25 cycles (30s on and 30s off). Sonicated tissues were centrifugated to obtain the supernatant. The supernatant was pre-cleared with Dynabeads Protein G beads (Invitrogen) for 2hrs at 4˚C. 5% of the pre-cleared sample was set aside as the Input, while the rest was incubated with 5μl anti-GFP antibody (A-11122, Invitrogen) overnight at 4˚C. The immune-precipitated (IP) sample was incubated with Dynabeads Protein G beads for 5hrs at 4˚C to allow beads binding. After that, beads were washed 5 times in LiCl Wash Buffer (10mM Tris pH7.4, 500mM LiCl, 1% NP-40, 1% sodium deoxycholate), while the Input sample was incubated with 1μl 10mg/ml RNase A at 37˚C for 1hr. Both IP and Input samples were incubated with 100μg proteinase K in PK Buffer (200mM Tris pH7.4, 25mM EDTA, 300mM NaCl, 2% SDS) first at 55˚C for 3hrs and then at 65˚C overnight. DNA was purified by phenol-chloroform extraction and the concentration was measured by Qubit. Four biological replicates of testis (*bam$^{A86/Df}$, GFP-Rhi*) ChIP were done and used in qPCR, with two randomly selected replicates sequenced. Two biological replicates of ovary ChIP (*GFP-Rhi*) were done and sequenced. ChIP-qPCR was normalized first to Input and then to a negative control region (free of Rhi binding in ovary according to Mohn et al. 2014) to obtain Rhi enrichment (primers listed below). ChIP DNA was subject to library prep using NEBNext ChIP-Seq Library Prep Master Mix Set for Illumina (NEB E6240) according to manufacturer's instructions. Two biological replicates per sex were sequenced on Illumina HiSeq 2500 yielding 13–22 million

50bp single-end reads. Reads were mapped to the genome as described in [20] permitting local repeats. Coverage over piRNA clusters were computed and the enrichment of IP over input was calculated. Two biological replicates were consistent with a correlation coefficient >0.96, so the average enrichment was plotted.

*42AB* [14]

F: 5'-GTG GAG TTT GGT GCA GAA GC

R: 5'-AGC CGT GCT TTA TGC TTT AC

*flam* [14]

F: 5'-TGA GGA ATG AAT CGC TTT GAA

R: 5'-TGG TGA AAT ACC AAA GTC TTG GGT CAA

*Su(Ste)* [31]

F: 5'-CTTGGACCGAACACTTTGAACCAAGTATT

R: 5'-GGCATGATTCACGCCCGATACAT

*AT-chX* [67]

F: 5'-AGCGATCCCACTGCTAAAGA

R: 5'-GTCGAAGACGTCCAGAGGAG

*negative control for Rhi binding* (this study)

F: 5'-AAGAGCAGAGGGGCCAAATC

R: 5'-TCCAAGTCGGCTTCCCTTTC

## Genome-wide relationship between piRNA production and Rhi binding

This analysis was adapted from Mohn et al. (2014) with modifications. piRNA production and Rhi enrichment were computed for individual 1Kb windows tiling the entire dm6 genome. The average of two well-correlative biological replicates was used for this analysis. Only 1Kb windows having both ≥4 RPM piRNAs in controls and ≥30 RPM reads in IP samples were plotted. Rhi-dependent loci (RD loci) were defined as 1Kb windows showing ≥4-fold drop in piRNA production in *rhi* mutant testes, and the rest were treated as Rhi-independent loci (RI loci). Local regression was implemented with LOESS technique in python.

## Data visualization and statistical analysis

Most data visualization and statistical analysis were done in Python 3 via JupyterLab using the following software packages: numpy [69], pandas [70] and altair [71]. Germ cell death, GSC loss and ChIP-qPCR were plotted in GraphPad Prism. Mann–Whitney–Wilcoxon test was done to compute p values for GSC loss. Multiple t-tests corrected for multiple comparisons by the Holm-Sidak method were done for Rhi ChIP-qPCR, using the uni-strand cluster *flam* known to be free of Rhi binding as a negative control. The UCSC Genome Brower [72] and IGV [73,74] were used to conduct explorative analysis of sequencing data.

## Supporting information

**S1 Fig. Del and Cuff are required for normal male fertility.** Compromised fertility *of del* (top) and *cuff* (bottom) mutant males. Sperm exhaustion test of *del^{HN/WK}* or *cuff^{WM25/QQ37}*

mutant (orange) and respective heterozygous sibling control (blue) males. Left: averaged numbers of offspring per male 1–11 days after eclosion (n = 5). Right: total number of progeny per male after mating for 11 days. Shaded areas and error bars display standard error. *P* value from unpaired t-test.
(TIF)

**S2 Fig. Rescue of female sterility by transgenes used in this study.** GFP-tagged Rhi, Del and Cuff transgenes as well as mKate2-tagged Rhi transgene driven by a putative *rhi* promoter fully rescue the female sterility of respective mutations.
(TIF)

**S3 Fig. Characterization of the expression of Rhi in testis.** Individual channels of images shown in Fig 2D.
(TIF)

**S4 Fig. Moon acts downstream of Rhi in testis.** Confocal images of apical tips of testes expressing GFP-Rhi, in *moon*$^{\Delta1}$ and *moon*$^{\Delta28}$ mutant backgrounds. Note that *moon* is X-linked, so XY males only have one copy of *moon* and trans-heterozygous mutant cannot be generated.
(TIF)

**S5 Fig. Production of TE-antisense and TE-sense piRNAs collapses without *rhi*.** Loss of TE-mapping piRNAs in testes of *rhi* mutants. Scatter plot showing expression of TE-antisense (left) and TE-sense (right) piRNAs in *rhi*$^{2/KG}$ mutant versus heterozygous control testes. piRNAs that show ≥2-fold reduction (FDR < 0.05) and ≥10 RPM average expression levels are marked in blue. Shown are averages of two biological replicates.
(TIF)

## Acknowledgments

We are grateful to Xin Chen, Peter Andersen, William Theurkauf, Trudi Schüpbach, Paul Lasko, Elena Pasyukova and three Drosophila Stock Centers (Bloomington, Vienna, Kyoto) for fly stocks. We thank Katalin Fejes Toth and members of Aravin lab for discussion and comments. We appreciate the help of Maria Ninova and Fan Gao (Bioinformatics Resource Center, Caltech) with bioinformatics analysis, the help of Grace Shin and Maayan Schwarzkopf with HCR experiments, the help of Giada Spigolon and Andres Collazo (Biological Imaging Facility, Caltech) with microscopy, and the help of Igor Antoshechkin (Millard and Muriel Jacobs Genetics and Genomics Laboratory, Caltech) with sequencing.

## Author Contributions

**Conceptualization:** Peiwei Chen, Alexei A. Aravin.

**Data curation:** Peiwei Chen, Alexei A. Aravin.

**Formal analysis:** Peiwei Chen, Alexei A. Aravin.

**Funding acquisition:** Alexei A. Aravin.

**Investigation:** Peiwei Chen, Yicheng Luo.

**Methodology:** Peiwei Chen, Alexei A. Aravin.

**Project administration:** Peiwei Chen, Alexei A. Aravin.

**Resources:** Peiwei Chen, Alexei A. Aravin.

**Software:** Peiwei Chen, Alexei A. Aravin.

**Supervision:** Peiwei Chen, Alexei A. Aravin.

**Validation:** Peiwei Chen, Yicheng Luo, Alexei A. Aravin.

**Visualization:** Peiwei Chen, Alexei A. Aravin.

**Writing – original draft:** Peiwei Chen, Alexei A. Aravin.

**Writing – review & editing:** Peiwei Chen, Alexei A. Aravin.

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
