## [Decision Letter · Decision Letter 0]

9 Dec 2020

Dear Dr Aravin,

Thank you very much for submitting your Research Article entitled “RDC complex executes a dynamic piRNA program during Drosophila spermatogenesis to safeguard male fertility” to PLOS Genetics. Your manuscript was fully evaluated at the editorial level and by three independent peer reviewers. The reviewers appreciated the attention to an important problem, but reviewers 1 and 3 raised important concerns about the current manuscript. Based on the reviews, we will not be able to accept the manuscript, at least in its current version. However, given the positive comments, we would be willing to review again a much-revised version including experimental work. We cannot, of course, promise publication at that time. The most important concerns are:

Experimental demonstration that Rhino only localizes on AT-ChX at later stages of spermatocytes development.Extend the characterization of Rhino mutant germ cells to Deadlock and Cuff, as pointed out by reviewer 3. We do not ask to repeat all the experiments with Deadkock and Cuff mutants, but for example tests such as the SEA would be important in the revised manuscript.

We feel that these experiments should be performed before re-submission.

Should you decide to revise the manuscript for further consideration here, your revisions should address all the remaining points made by each reviewer. We will also require a detailed list of your responses to the review comments and a description of the changes you have made in the manuscript.

If you decide to revise the manuscript for further consideration at PLOS Genetics, we would appreciate an expected resubmission date by email to plosgenetics@plos.org.

[LINK]

We are sorry that we cannot be more positive about your manuscript at this stage. Please do not hesitate to contact us if you have any concerns or questions.

Yours sincerely,

Jean-René Huynh

Associate Editor

PLOS Genetics

Gregory P. Copenhaver

Editor-in-Chief

PLOS Genetics

Reviewer's Responses to Questions

**Comments to the Authors:**

Reviewer #1: Uploaded as an attachment

Reviewer #2: In this manuscript, Chen et al. explores the function of RDC (Rhino-Deadlock-Cutoff) complex in Drosophila testis. RDC complex mediates transcription of piRNA cluster, and has been believed to function only in female germline. However, Chen et al. provide a convincing evidence that RDC complex has a function in male germline as well, by showing compromised fertility of male RDC mutants. They further characterize targets of RDC complex in the testis, in comparison to ovary, providing insights into the regulation and biogenesis of piRNA in Drosophila testis. This is an important study to clarify the function of piRNA pathway components in the testis via thorough analysis, and is a significant contribution to the field.

I have no major issues with this manuscript, with only minor suggestions as following.

-Fig3A, ‘no germline’ is not entirely accurate. I see elongated sperm tails (as the haze of green background). It should be labeled as ‘early germ cell depletion’.

-The passage in the discussion ‘Also, after transition to spermatocytes, germ cells lose their ability to de-differentiate back to germline stem cells (Brawley and Matunis, 2004). It is thus tempting to propose that piRNA pathway contributes to the maintenance of cellular plasticity in early male germ cells (GSCs, gonialblasts and spermatogonia) by ensuring robust genome defense.’: there is no evidence to go against this statement, but it is generally thought that pre-meiotic S phase would prepare chromosome/chromatids to be compatible with meiosis (consecutive homolog segregation and sister segregation without intervening S phase), and therefore, it is highly unlikely that they can resume mitotic divisions. Therefore, the comments to suggest piRNA might maintain germ cell plasticity, merely basing on the expression pattern of piRNA, seems to be extrapolation. This is a discussion, so the author should be allowed to say almost anything they like, but I think this discussion might rather hurt authors for putting a rather careless discussion. So I am just pointing it out.

Reviewer #3: This manuscript from Chen et al, revealed the role of Rhino and the RDC complex in piRNA biogenesis from dual strand piRNA clusters in testis all along spermatogenesis. The role of Rhino was already well characterized in piRNA biogenesis in ovaries but due to a lack of mutant phenotype in males was remain largely unexplored in testis. A detail examination of the male fertility by sperm exhaustion test revealed subfertility of the Rhino mutant males. A spatio temporal careful analysis indeed revealed that the RDC complex is not ovary-specific and assembled also in early spermatogenesis which is required for piRNA production and TE silencing. Surprisingly and interestingly the piRNA biogenesis of suppresser of stellate piRNAs which is expressed from a particular dual strand cluster is not Rhino dependent.

The manuscript is well written and fairly easy to follow. The data are mostly well presented and clearly explained. The main criticism is that all along the manuscript the authors conclude on the effect of a loss of the RDC complex while they investigate, most of the time, only the loss of Rhino. All the experiments presented in this paper (small RNA sequencing, copia LacZ staining, TE mRNA expression …) are not investigated in del or cuff mutant. Overall, this is an interesting paper that is suitable for publication in Plos Genetics, as long as the following minor criticisms can be addressed.

Minor points

Figure 1: The phenotype of the Rhino is very well described and it is clear that Rhino is required for male fertility. Are Deadlock and Cutoff required like Rhino for male fertility?

Figure 2 B: Rhino is dispersed in the nuclei and not puncta anymore in del or cuff mutants. However, in del or cuff KD ovaries the protein Rhino is completely lost as demonstrated in the paper of ElMaghraby et all. The difference between testis and ovaries should be discussed.

Figure 2D: Only the merge is presented. For a better visualization and interpretation an independent staining of each proteins should be added to this figure.

Figure 2A: The expression of Moon appears to be more diffused in the nucleus in the GSC and then punctuated in the spermatogonia. It is not obvious that Moonshiner colocalize with Rhino in the GSC. It should be mentioned and discussed.

From the experiment presented figure 3, the authors should conclude that the loss of Rhino instead of RDC complex leads to a reduction in the germ cell count and GSC population in testis since they have only investigated Rhino mutant.

Only the antisense piRNAs which are the regulatory ones are presented. Since Rhino is involved in dual strand piRNA cluster expression it would be nice if the authors could also present the effect of a loss of Rhino on sense piRNAs.

The interpretation of the figure 4A is that Rhino and not the RDC complex is essential for piRNA production from a large fraction of piRNA clusters in male germline.

Do the authors know whether TEs contained in the 38C piRNA cluster are TEs which are specifically expressed in testis when they are derepressed ? What is the size of the 38C cluster?

Again, the conclusion of ChIP experiments should be that Rhino and not the RDC complex binds the chromatin and ensures the expression of Dual strand piRNA-clusters.

To explain the specific localization at one single locus of Rhino, the authors proposed that since the germ cells are diploid it is plausible that this locus resides on one of the two sets chromosomes. It is well described in drosophila cells that the two sets of autosomes are pairs together and form only one dot in the nucleus in FISH. In ovaries the pairing originates immediately after the stem cell stage. This pairing occurs well before the initiation of meiosis and, strikingly, continues through the several mitotic divisions preceding meiosis. Is it not the case in male germ cells?

To prove the specificity of the situ HCR experiment performed to detect nascent transcripts of the 38C locus the experiment should also be performed in a Rhino mutant genetic background.

**Have all data underlying the figures and results presented in the manuscript been provided?**

Reviewer #1: Yes

Reviewer #2: Yes

Reviewer #3: Yes

PLOS authors have the option to publish the peer review history of their article (what does this mean?). If published, this will include your full peer review and any attached files.

Reviewer #1: **Yes: **Phillip D. Zamore

Reviewer #2: **Yes: **Yukiko Yamashita

Reviewer #3: No

---

## [Decision Letter · Decision Letter 1]

10 May 2021

Dear Dr Aravin,

We are pleased to inform you that your manuscript entitled "RDC complex executes a dynamic piRNA program during Drosophila spermatogenesis to safeguard male fertility" has been editorially accepted for publication in PLOS Genetics. Congratulations!

Yours sincerely,

Jean-René Huynh

Associate Editor

PLOS Genetics

Gregory P. Copenhaver

Editor-in-Chief

PLOS Genetics

Comments from the reviewers (if applicable):

Reviewer's Responses to Questions

**Comments to the Authors:**

Reviewer #1: The authors have done a superb job revising the manuscript. I enthusiastically recommend it for publication in PLoS Genetics.

Reviewer #2: I do appreciate that the authors took a great care to address reviewer comments. My comments were certainly addressed, and also they did a great job addressing other reviewers' comments as well. I also appreciate reviewer #1's thorough comments as well, which made the manuscript much stronger. There may be other additional issues that could still make the manuscript better, but I am of the opinion that if the manuscript's main conclusion/message is well supported, other issues may be left open (as far as the authors do not make conclusions that cannot be made). Overall, this is an important study that provide new analysis on piRNA mechanism in the Drosophila testis.

Reviewer #3: The authors have addressed all of my concerns and the current version is acceptable for publication.

**Have all data underlying the figures and results presented in the manuscript been provided?**

Reviewer #1: Yes

Reviewer #2: Yes

Reviewer #3: Yes

PLOS authors have the option to publish the peer review history of their article (what does this mean?). If published, this will include your full peer review and any attached files.

Reviewer #1: No

Reviewer #2: **Yes: **Yukiko Yamashita

Reviewer #3: No

**Data Deposition**

http://datadryad.org/submit?journalID=pgenetics&manu=PGENETICS-D-20-01652R1

**Press Queries**

---

## [Editor Report · Acceptance letter]

9 Jul 2021

PGENETICS-D-20-01652R1 

RDC complex executes a dynamic piRNA program during Drosophila spermatogenesis to safeguard male fertility 

Dear Dr Aravin, 

We are pleased to inform you that your manuscript entitled "RDC complex executes a dynamic piRNA program during Drosophila spermatogenesis to safeguard male fertility" has been formally accepted for publication in PLOS Genetics! Your manuscript is now with our production department and you will be notified of the publication date in due course.

With kind regards,

Olena Szabo

PLOS Genetics

On behalf of:
